# AN ADVERSARIAL ATTACK VIA FEATURE CONTRIBUTIVE REGIONS

## ABSTRACT

Recently, to deal with the vulnerability to generate examples of CNNs, there are many advanced algorithms that have been proposed. These algorithms focus on modifying global pixels directly with small perturbations, and some work involves modifying local pixels. However, the global attacks have the problem of perturbations' redundancy and the local attacks are not effective. To overcome this challenge, we achieve a trade-off between the perturbation power and the number of perturbed pixels in this paper. The key idea is to find the feature contributive regions (FCRs) of the images. Furthermore, in order to create an adversarial example similar to the corresponding clean image as much as possible, we redefine a loss function as the objective function of the optimization in this paper and then using gradient descent optimization algorithm to find the efficient perturbations. Our comprehensive experiments demonstrate that FCRs attack shows strong attack ability in both white-box and black-box settings for both CIFAR-10 and ILSVRC2012 datasets.

## 1 INTRODUCTION

The development of deep learning technology has promoted the successful application of deep neural networks (DNNs) in various fields, such as image classification (Krizhevsky et al., 2012; Simonyan & Zisserman, 2014), computer vision (He et al., 2016; Taigman et al., 2014), natural language processing (Devlin et al., 2018; Goldberg, 2017), etc. In particular, convolutional neural networks (CNNs), a typical DNNs, have shown excellent performance applied in image classification. However, many works have shown that CNNs are extremely vulnerable to adversarial examples (Szegedy et al., 2013). The adversarial example is crafted from clean example added by well-designed perturbations that are almost imperceptible to human vision, while can fool CNNs. Scholars have proposed a variety of methods to craft adversarial samples, such as L-BFGS (Szegedy et al., 2013), FGSM (Goodfellow et al., 2014), I-FGSM (Kurakin et al., 2016), PGD (Madry et al., 2017) and C&W (Carlini & Wagner, 2017). These attack strategies can successfully mislead CNNs to make incorrect predictions, restricting the application of CNNs in certain security-sensitive areas (such as autonomous driving, financial payments based on face recognition, etc.). Therefore, learning how to generate adversarial examples is of great significance.

We can categorize these attacks into two categories, i.e., the global attacks and the local attacks, according to the region added perturbations. The global attacks tempt to perturb all pixels of the clean image, such as FGSM (Goodfellow et al., 2014), PGD (Madry et al., 2017) and C&W (Carlini & Wagner, 2017); the local attacks only modify some pixels of the clean image, such as one-pixel attacks (Su et al., 2019) and JSMA (Papernot et al., 2016b). At present, the global attacks perturb all pixels on the whole image, which not only fail to destroy the feature contributive regions (the critical semantics of an image), but they also increase the degree of image distortion. We explain in detail in the experimental part. The local attacks seem to be able to solve this problem, but the current proposed local attacks don't well realize that focus on undermining the image feature contributive regions. Papernot et al. (2016b) proposed a method of crafting adversarial example based on the Jacobian Saliency Map by constraining the $\ell_0$ norm of the perturbations, which means that only a few pixels in the image are modified. However, this method has the disadvantage of over-modifying the value of the pixels, making the added perturbations easily perceptible by the naked eye, and its adversarial strength is weak (Akhtar & Mian, 2018). Su et al. (2019) proposed an extremely adversarial attack—one-pixel attack. One-pixel attack can fool CNNs by changing 1 to 5 pixels, but

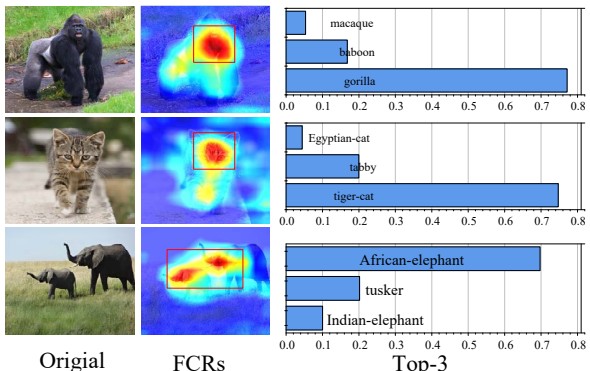

Origial     FCRs     Top-3

Figure 1: We use Grad-CAM to get the heatmap of the image and the red frame area is the feature contribution regions involved in this work.

this method is better for low-resolution images attack (such as CIFAR-10), and the attack success rate for high-resolution images will be greatly reduced (such as ImageNet), and the cost is very large $\ell_1$ distortion (Xu et al., 2018).

In this paper, we propose a novel attack method to overcome the redundant perturbations of the global attacks and the poor strength of the proposed local attacks. Inspired by the work of CAM (Zhou et al., 2016) and Grad-CAM (Selvaraju et al., 2017), it is the most effective way to reduce image distortion, high efficiency and reduce computational complexity by adding perturbations to the critical semantics. As we all know, CNN is an end-to-end representation learning model, which starts from simple low-level features and combines them into abstract high-level features layer by layer. Thus, Grad-CAM (Selvaraju et al., 2017) uses the gradient information of the last convolutional layer as the metric to understand the decision of each neuron for target classification, and explains in a visual way that not all image pixels contribute to the model classification. Similarly, as shown in Figure 1, the red area is the main contributive area. Therefore, perturbing the image globally is not the most efficient strategy. We propose the FCRs attack strategy, which only adds perturbations in Feature Contributive Regions (FCRs) with the aim of generating sparse and more excellent perturbations. Especially, compared with existing local attacks, our proposed method perturbs continuous semantic regions rather than discrete pixels. In this work, we use Grad-CAM to locate regions that have a greater impact on the classification decision of CNNs. To ensure the similarity between the adversarial example and the corresponding clean image as much as possible, the objective function we optimize is the sum of the two parts of the function: the $\ell_2$ norm of the perturbations and the loss function of the generated adversarial examples. We thus use the stochastic gradient descent optimization algorithm to find efficient perturbations. In order to avoid the situation where the perturbations do not update when the objective function tends to zero, we also introduce inverse temperature $T$ under the inspiration of Hinton et al. (2015).

Compared to previous work, the contributions of our work are summarized as follows:

- We propose an attack via feature contributive regions (FCRs) for achieving a trade-off between the powerful attack and the small perturbations. More importantly, this work implements an effective local attack algorithm by redefining an objective function.

- Specially, we novelly propose an inverse temperature $T$, which avoids the situation where the loss function of the generated adversarial example tends to be zero when the stochastic gradient descent optimization algorithm is used to find the perturbations.

- Comprehensive experiments demonstrate that FCRs attack consistently outperforms state-of-the-art methods on the CIFAR-10 and ILSVRC-2012 datasets. In addition, we verify the importance of FCRs by dividing the original clean image into two parts (i.e., FCRs and Non-FCRs).

## 2 RELATED WORK

In many cases, the CNNs are vulnerable to adversarial attacks which have caused extensive research in academia. Szegedy et al. (2013) used the constrained L-BFGS algorithm to craft adversarial ex-

 amples. L-BFGS attack has a high attack success rate, but the computational cost is also high (Narodytska & Kasiviswanathan, 2017). Therefore, Goodfellow et al. (2014) proposed FGSM, which can quickly generate adversarial examples but has a low attack success rate. Kurakin et al. (2016) proposed the Iterative attack method (I-FGSM) on the basis of FGSM and Madry et al. (2017) proposed PGD. Dong et al. (2018) proposed an iterative algorithm based on momentum (MI-FGSM) to improve the transferability of adversarial samples. Xie et al. (2019) combined the input diversity strategy with iterative attacks on I-FGSM and MI-FGSM to further improve the transferability of adversarial examples. The aforementioned attacks belong to the gradient attack family, and they destroy the semantic information of the whole image. Papernot et al. (2016b) proposed an attack method based on the Jacobian Saliency Map by minimizing the $\ell_0$ norm of adversarial perturbations and used a greedy algorithm to find saliency pixels. However, this method has the problems of over-modifying pixels too much and weak attack intensity. Su et al. (2019) proposed an adversarial attack method based on the differential evolution algorithm. This method also focuses on the number of pixels to be modified, but does not limit the power of a single change, thus leading to very large $\ell_1$ distortion (Xu et al., 2018). In this work, we expect to achieve a more effective attack that can be as successful as existing attacks but achieves a trade-off between the perturbation power and the number of perturbed pixels. We will show that the proposed FCRs attack is able to destroy the feature contribution regions that make attacks successful, but without incurring extra pixel-level perturbations.

Related to our work is Deng & Zeng (2019), who proposed a spatial transformed attack method based on attention mechanism. This work expands the stadv (Xiao et al., 2018) to A-stadv. The purpose of this work is to generate adversarial examples with less interference and less visible. The author only conducts experiments on the ImageNet dataset, and does not discuss the black-box attack effect of this method. But while verifying that many pixel-level perturbations are redundant, our work proposes a new algorithm to craft perturbations, and demonstrates its white-box and black-box attack effects on the CIFAR-10 and ILSVRC2012 datasets. In addition, Xu et al. (2019) used CAM to explain adversarial perturbations but their target is not to generate adversarial examples, but to understand and interpret adversarial examples. Zhang et al. (2020) proposed a target-free method to generate adversarial examples via principal component analysis and made adversarial examples relate to the data manifold, but their experiment showed that the performances of their method were not always better than FGS and C&W. Here we pay more attention to the feature contribution regions and finally, we achieve a trade-off between the powerful attack and the number of perturbed pixels.

## 3 METHODOLOGY

Inspired by "attention mechanism" (Zagoruyko & Komodakis, 2016), we believe the classifier's performance is greatly affected by some specific feature regions that is termed as feature contributive regions (FCRs) in this paper. This intuition is also confirmed by Deng & Zeng (2019) proposed A-stadv which is an attention based on spatial transformed adversarial example. Therefore, if we find FCRs and add perturbations to them, it will be more effective to fool the classifier with fewer perturbations than previous methods. Our idea is to divide an image into two semantic parts: FCRs and Non-FCRs and then perturbs feature contributive regions. The result of fewer perturbations ensures maximumly adversarial effects on local regions of clean images.

### 3.1 NOTIONS

**Deep neural networks (DNNs):** A DNN can be expressed as a high-dimensional approximation function: $f(X, \theta) : \mathbb{R}^m \to \mathbb{R}^n$, where $X \in \mathbb{R}^m$ is the input variable, $Y \in \mathbb{R}^n$ is the true class, $X$ and $\theta$ represents the model parameters. In this work, we focus on a specific DNN, convolutional neural networks (CNNs) that are typically comprised of convolutional layers with some method of periodic downsampling (either through pooling or strided convolutions). Here, we define the Logits layer. The input before the softmax layer of the CNNs, namely the Logits layer (the penultimate layer): $Y_j = w_j^T A, j = 1, 2, \ldots, C$, where $w_j^T$ is the weight matrix and $A$ is the input vector of the Logits layer, which contains a mapping function $X \mapsto A$. Then the softmax function can be expressed as $S_j = \exp Y_j / \sum_{i=1}^c \exp Y_i$, and the final model can be expressed as $f(X) = S\left(w_j^T A\right)$. Given an input $X$, then the predicted class of $X$ can be expressed as $\hat{Y} = \arg\max_{j=1,\ldots,k} f(X)_j$. The goal

of model training is to minimize the cross-entropy loss function, which can be expressed as:

$$J = -\sum_{j=1}^{C} Y_j \log S_j = -\log S_j \tag{1}$$

where $Y$ is a $1 \times C$ vector and there are $C$ values in it. Only one value is 1 (corresponding to the true label), and the other $C - 1$ values are all 0. For $N$ input-label pairs $(X_i, Y_i)$, the cross-entropy loss function of the model can be expressed as:

$$J = -\frac{1}{N} \sum_{i=1}^{N} \sum_{j=1}^{C} Y_j \log S_j = -\frac{1}{N} \sum_{i=1}^{N} \log S_j \tag{2}$$

**Adversarial examples:** An adversarial example can be represented as $X' = X + \delta$, where $\delta$ is the perturbation. Normally, the perturbation $\delta$ is constrained by the $\ell_0$, $\ell_2$ or $\ell_\infty$ norm, that is $\|X' - X\|_p \leq \epsilon$. For untargeted attacks, we only need to search for an $X'$ satisfying $Y' = \arg\max_j f(X')_j$, where $Y' \neq Y$ and we also do not need to specify which class will be misclassified; for targeted attacks, we specify a target class $Y^* \neq Y$, so that the target model not only misclassifies the example, but also needs to classify them into the specified class. In general, the targeted attacks are more difficult than untargeted attacks.

### 3.2 Feature Contributive Regions (FCRs)

FCRs refer to the regions in an image that are critical for model prediction. We can utilize Grad-CAM (Selvaraju et al., 2017), CAM (Zhou et al., 2016) and c-MWP (Zhang et al., 2018) to observe FCRs. However, compared with CAM and c-MWP, Grad-CAM is not restricted by a specific CNNs architecture. In addition, it can generate better quantitative and qualitative results with less computation. As a result, we use Grad-CAM to search for FCRs in our work.

Suppose the input image $X$ is forward propagated through the CNNs, and the last layer of convolutional layer outputs the high-level feature map $A$ of the image, where $A^k \in \mathbb{R}^{n \times v}$ represents the activation of the $k$-$th$ convolution kernel with the size of $u \times v$. $A$ outputs the score vector $Y$ (also called logits) of each class after passing through a fully connected layer FC, where $Y^C$ represents the logits value of the $C$-$th$ class. To this end, we compute the gradient of $Y^C$ to $A^k$, i.e. $\partial Y^C / \partial A^k$ to measure the classification prediction importance of the $k$-$th$ convolution kernel to the $C$-$th$ class. Furthermore, we adopt the global average pooling operation to calculate the weight $\lambda_k^C$ of the $k$-$th$ convolution kernel:

$$\lambda_k^C = \frac{1}{Z} \sum_i \sum_j \frac{\partial Y^C}{\partial A_{ij}^k} \tag{3}$$

where $Z = u \times v$, $A_{ij}^k$ is the activation at cell $(i, j)$ of the $k$-$th$ convolution kernel. We use the weight $\lambda_k^c$ to perform a weighted summation of $A^k$, and calculate a feature activation map $\sum_k \lambda_k^C A^k$ for the $C$-$th$ class. Considering that only the positive value in $\sum_k \lambda_k^C A^k$ will have a positive effect on the final classification result, the final weighted result is reactivated by ReLU to remove the influence of the negative value, and the activation map of the $C$-$th$ class is obtained:

$$L_X = \mathrm{ReLU}\left(\sum_k \lambda_k^C A^k\right) \tag{4}$$

We can visualize $L_X$ in the form of heatmap (e.g. Figure 1), in which the red area is the feature contribution regions (FCRs) to the $C$-$th$ class.

Since the FCRs are usually irregular, we introduce a masking mechanism to locate. Formally, the mask is a 0-1 matrix with the same size of the input image. The element is 0 in $mask_X$ indicates the corresponding pixel in the image is not in the FCRs. On the contrary, the element is 1 indicates the corresponding pixel is in the FCRs. Thus, we can obtain the FCRs of the image by simply Hadamard product applied between the mask and the image. For obtaining the mask, a simple threshold mechanism can be utilized:

$$\mathrm{mask}_X = \begin{cases} 1 & [L_X] \geq t \\ 0 & \mathrm{others} \end{cases} \tag{5}$$

where $t$ is a threshold and $L_X$ indicates that the input image $X$ is the $C$-$th$ class activation map. Our proposed method uses $mask_X$ to locate the location of the added perturbations $\delta_{FCR}$.

### 3.3 Generate Perturbations for FCRs

We now turn to our approach to generate adversarial perturbations. To begin, we rely on the initial formulation of adversarial perturbations (Goodfellow et al., 2014) and formally define the problem as follows:

$$\min_{\delta} \quad \|\delta\|_p \tag{6}$$
$$\text{s.t.} \quad f(X + \delta) \neq y,$$
$$X + \delta \in [0, 1]^m.$$

where $\|\cdot\|_p$ is the norm that constrains perturbations $\delta$. The commonly used $p$-norm is $\ell_0$, $\ell_2$ or $\ell_\infty$. $X$ is fixed, and the goal is to find the minimal $\delta$ that can fool the CNNs.

Our method is different in that only perturbs FCRs, so we solve this problem by formulating it as follows:

$$\min_{\delta_{FCR}} \quad \|\delta_{FCR}\|_p \tag{7}$$
$$\text{s.t.} \quad f(X + \delta_{FCR}) \neq y,$$
$$X + \delta_{FCR} \in [0, 1]^m.$$

However, the exact and direct computation of $\|\delta_{FCR}\|_p$ is difficult for existing algorithms, as the constraint $f(X + \delta) \neq y$ is highly non-linear. Therefore, we approximate $\|\delta_{FCR}\|_p$ in a different form that is better suited for optimization. We define an objective function $F$ satisfying $f(X + \delta_{FCR}) \neq y$. This objective function consists of two parts: (1) a loss function for generating adversarial examples, and (2) an $\ell_2$ regularization function to limit the perturbations. In theory, the $\ell_0$ and $\ell_\infty$ norms can also be considered as a regularization function. However, we notice that the $\ell_0$ norm is non-differentiable and cannot be adopted for the standard gradient descent algorithm. In addition, the $\ell_\infty$ norm only focuses on the largest value in $\delta_{FCR}$, it is easy to oscillate between the two suboptimal solutions during the gradient descent process (Carlini & Wagner, 2017). Therefore, we use the $\ell_2$ norm of the perturbations $\delta_{FCR}$ as the distance metric. Thus we define the objective function as follows:

$$F = \beta \frac{1}{\|\delta_{FCR}\|_2} + J\left(f_\theta\left(X + \delta_{FCR}\right), Y\right) \tag{8}$$

where $\beta$ is a hyper-parameter that controls the degree of distortion. For the clean image $X$, our optimization goal is to find the $\delta_{FCR}$ that maximizes the objective function $F$ when the model is misclassified:

$$\max_{\delta_{FCR}} \quad F \tag{9}$$
$$\text{s.t.} \quad X + \delta_{FCR} \in [0, 1]^m.$$

Since maximizing $F$ and minimizing $1/F$ are equivalent, we can get the following optimization problem:

$$\min_{\delta_{FCR}} \quad 1/F \tag{10}$$
$$\text{s.t.} \quad X + \delta_{FCR} \in [0, 1]^m.$$

Then we use the stochastic gradient descent (SGD) algorithm to solve the $\delta_{FCR}$. The gradient of $1/F$ in $\delta_{FCR}$ is $\nabla_{\delta_{FCR}}(1/F)$ and it is used to update $\delta_{FCR}$ iteratively:

$$\delta_{\text{FCR}} = (\delta_{FCR} - \nabla_{\delta_{FCR}}(1/Loss) \times LR) \odot \text{mask}_X \tag{11}$$

where $LR$ is a hyper-parameter, which is equivalent to the learning rate.

Firstly, we generate a random perturbation $\delta_{FCR}$, and get the initial adversarial example $X' = X + \delta_{FCR}$. From Eq. (1) we can know that when $S_j \to 1$, $J_{adv} \to 0$, we set $P = \beta\left(1/\|\delta_{\text{FCR}}\|_2\right)$, $J_{adv} = J\left(f_\theta\left(X + \delta_{FCR}\right), Y\right)$. At this time $1/F = 1/P$, $\nabla_{\delta_{FCR}}(1/F) = \nabla_{\delta_{FCR}}(1/P)$, then continue to use the stochastic gradient descent (SGD) algorithm to update $\delta_{FCR}$ will not lead to $J_{adv}$ becoming bigger. In order to avoid this situation, we use the distillation idea to introduce the hyper-parameter $T$ ($T \leq 1$). Applying $T$ will make $J_{adv}$ increase $\log T$, then Eq. (1) becomes the following form:

$$J_{adv}^T = -\log\left(S_j/T\right) \tag{12}$$

Thus our objective function is modified to:

$$F = \beta \frac{1}{\|\delta_{\text{FCR}}\|_2} + J_{adv}^T \qquad (13)$$

---

**Algorithm 1** Generate Adversarial Examples via FCRs

---

**Input:** A clean image $X$; the iterations $N$; the learning rate $LR$; the degree of distortion $\beta$; the threshold $t$; the inverse temperature $T$

**Output:** $\delta_{FCR}$

1: initialize $\delta_{FCR}$
    // $K$ is the number of feature maps in the last layer of convolution layers
2: $\lambda_k^C \leftarrow \frac{1}{Z} \sum_p \sum_q \frac{\partial Y^C}{\partial A_{pq}^k}, k = 1, \ldots, K$
3: $L_X \leftarrow \text{ReLU}\left(\sum_k \lambda_k^C A^k\right)$
4: $\text{mask}_X = \begin{cases} 1 & [L_X] \geq t \\ 0 & \text{others} \end{cases}$   // Get FCRs
5: **for** $i = 1, \ldots, N$ **do**
6:     $X' \leftarrow X + \delta_{FCR}$
7:     $1/F \leftarrow 1/\left(P + J_{adv}^T\right)$
8:     $\delta_{FCR} \leftarrow (\delta_{FCR} - \nabla_{\delta_{FCR}}(1/F) \times LR) \odot \text{mask}_X$ // Update $\delta_{FCR}$
9: **end for**

---

## 4 EXPERIMENTS

We verify the proposed method in Section 3 by experiments: (1) FCRs is an important basis for the final classification decision; (2) FCRs attack will produce less perturbations and reduce the pixel search space; (3) In this section, we show the experimental results of white-box attack and black-box attack, which shows that FCRs attack has powerful white-box attack capability and high transferability.

### 4.1 EXPERIMENT SETUP

**Datasets and Models:** We validate our method on two benchmark datasets: CIFAR-10 (Krizhevsky et al., 2009) and ILSVRC2012 (Russakovsky et al., 2015). The CIFAR-10 consists of 60,000 images sized , including 10 categories, each with 6,000 images. There are 50,000 images for training and 10,000 for testing. The ILSVRC2012 image classification dataset contains 1,200 thousand images from 1,000 categories, and 50,000 images are used as the validation set. There is no point in attacking images that have been misclassified, so the images we use to generate adversarial examples are all images that are correctly classified by all models. We use VGG (Simonyan & Zisserman, 2014) and ResNet (He et al., 2016) series models on the two datasets.

**Evaluation indicators:** The evaluation indicators setting in this article are the attack success rate (ASR), the image quality assessment index—peak signal-to-noise ratio (PSNR) (Hore & Ziou, 2010) and the $\ell_2$ distortion of perturbations. In an ideal situation, we need to conduct stronger attacks with smaller perturbations, so that a higher PSNR and smaller $\ell_2$ distortion can be guaranteed.

### 4.2 VALIDATE THE IMPORTANCE OF FCRS

This section uses VGG and ResNet model structure to conduct experiments on CIFAR-10 to further illustrate that FCRs are the basis for model classification. We divide into FCRs and Non-FCRs by using (Figure 2(a)). The accuracy rate of the input FCRs is up to 85% and above. However, the accuracy rate of the input Non-FCRs is very low (Figure 2(b)). The experimental results show that FCRs have the greatest semantics to model decision-making and are the areas that have a positive contribution to model classification.

In order to show that global attacks will produce not powerful perturbations and FCRs adding perturbations is the most efficient way, we improve the FGSM algorithm and add different perturbations to FCRs and Non-FCRs (Appendix A). We conduct experiments on the CIFAR-10. We add different perturbations on FCRs and Non-FCRs for comparison and the results are summarized in Appendix A. Obviously, the perturbations only in FCRs hardly reduce the attack success rate, that say, the FCRs are the best areas to optimize perturbations in the optimization landscape.

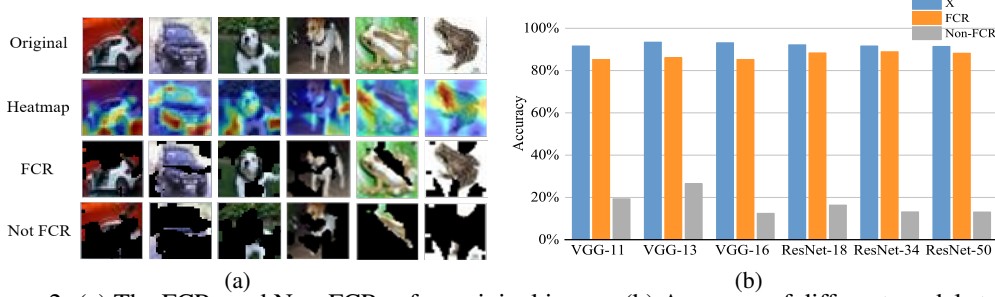

Figure 2: (a) The FCRs and Non-FCRs of an original image; (b) Accuracy of different model structures. For the $mask_X$, we use the threshold $t = 0.2$.

Table 1: Third column: Accuracy of the clean images on different models; Fourth column: ASR of FCRs attack to generate adversarial examples. (Hyper-parameter settings: $t = 0.2$, $T = 0.03$, $LR = 10$, $N = 20$, $\beta = 1$ and $t = 0.2$, $T = 0.05$, $LR = 20$, $N = 20$, $\beta = 1$ on the two datasets, respectively. The introduction of parameters is detailed in the Appendix B.)

| Datasets | Model | Accuracy | ASR |
|---|---|---|---|
| CIFAR-10 | VGG-11 | 91.66% | 99.70% |
| | ResNet-18 | 92.19% | 100.00% |
| ILSVRC2012 | VGG-19 | 71.02% | 100.00% |
| | ResNet-34 | 72.17% | 99.13% |

### 4.3 FCRs Attack

We generate adversarial examples on two datasets under the white-box setting. The results in Table 1 show the classification accuracy of the clean test data and the ASR of the adversarial examples generated by FCRs attack on different models. Figure 3 shows the perturbations and adversarial examples generated by the global attacks and FCRs attack. These are randomly selected from the examples that can be successfully attacked. It can be seen that the FCRs attack not only generates perturbations in the FCRs but also the adversarial examples are very close to the corresponding images. However, the images of global attacks are distorted greatly. When we use the same constraint of the $\ell_2$ distortion, we observe that the ASR of PGD is 74.33% and 56.50% on the two datasets; the ASR of C&W is 72.11% and 45.00%. In contrast, FCRs attack can still have powerful attack performance when it only attacks the local semantics.

### 4.4 Comparison with other methods

Table 2 reports the ASR, PSNR, and $\ell_2$ distortion of different attack methods (it is pointed out here that we are giving the average difference between the adversarial examples and the clean images). We show that the FCRs attack not only generates small perturbations (smaller $\ell_2$ distortion) but also has powerful attack performance (higher ASR), and the crafted adversarial examples are more similar to the original images (larger PSNR). Specifically, the distortion performance of C&W is the worst, that $\ell_2$ distortion is the largest on the two datasets, and the PSNR is also the smallest. Given that JSMA and one-pixel attack are both local attacks, we do a comparative experiment with

Table 2: ASR, PSNR and $\ell_2$ distortion for various attacks

| Datasets | Attack Methods | ASR | PSNR | $\ell_2$ |
|---|---|---|---|---|
| CIFAR-10 | PGD | 93.18% | 68.03 | 5.57 |
| | C&W | 97.44% | 57.09 | 21.01 |
| | JSMA | 90.33% | 60.12 | 19.53 |
| | One-pixel | 80.77% | 65.7 | 7.03 |
| | Ours | **100.00%** | **79.26** | **1.56** |
| ILSVRC2012 | PGD | 97.70% | 49.69 | 373.95 |
| | C&W | 99.33% | 46.86 | 543.76 |
| | JSMA | 90.00% | 60.08 | 60.64 |
| | One-pixel | 40.56% | **80.74** | **9.55** |
| | Ours | **100.00%** | 72.67 | 42.94 |

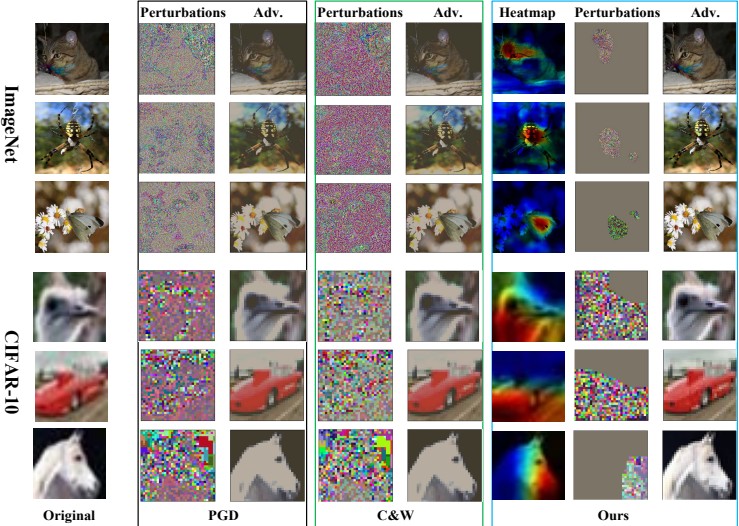

Figure 3: Comparison of perturbations and adversarial examples crafted by PGD, C&W and FCRs attack on the two datasets. Obviously, our proposed method only destroys critical semantics and reduces image distortion. (Our constraint on the $\ell_2$ norm of the three methods is 2 and 45.)

these two methods. On the CIFAR-10, the performance of JSMA is lower than the FCRs attack (ASR: 90.33% vs 100.00%); and its $\ell_2$ distortion is very large. On the ILSVRC2012, our method outperforms it in all metrics. We choose to attack 5 pixels for the one-pixel attack. On the CIFAR-10, one-pixel attack is not only large in $\ell_2$ distortion, but also has poor attack performance. On the ILSVRC2012, although the $\ell_2$ distortion of the one-pixel attack is the smallest, its attack success rate is only 40.56% and we observe that the one-pixel attack requires a lot of memory during the experiments. Thus we observe that the reduction of attack semantics does not reduce the performance of FCRs attack.

## 4.5 BLACK-BOX ATTACK

In this section, we explore a more challenging black-box scenario where the attacker first specifies an alternative model of the black-box model, and then generates a set of adversarial examples that can successfully attack the alternative model. Normally, this set of adversarial examples is considered to have strong transferability, that is, in the case of misleading alternative model, it will also mislead the target model (Papernot et al., 2016a). The underlying assumption is that highly transferable adversarial examples can achieve similar attack performance on many different target models (Papernot et al., 2017). Therefore, we can expect that transferable adversarial samples will reduce the accuracy of the alternative model and at the same time reduce the accuracy of the target model, resulting in high black-box attack capabilities. In order to prove the black-box attack capability of the FCRs attack, we conduct black-box attack experiments on different target models and datasets. As shown in Table 4 and 5, the adversarial examples generated by FCRs attack is more transferable in most cases.

## 5 CONCLUSIONS

This work explores the method of generating perturbations via the feature contribution regions. This article provides evidence to prove that the attack on the local semanticsis is the most effective. As our theory and experiments have shown, we have devised a more excellent attack method. We conduct extensive experiments with the CIFAR-10 and ILSVRC2012 datasets. The results show that FCRs attack are much stronger than existing global attacks (such as PGD and C&W) and local attacks (such as JSMA and One-Pixel), and the attack based on feature contribution regions may also provide a new perspective for future research on better defensive methods.

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

# A   IMPROVING FGSM VIA FCRS

Here, we give the algorithm of combining FGSM and FCRs.

---

**Algorithm 2** FGSM + $mask_X$

---

**Input:** A clean image $X$; Perturbations $\epsilon_1$ of FCRs; Perturbations $\epsilon_2$ of Non-FCRs
**Output:** $X'$

1: $\lambda_k^C \leftarrow \frac{1}{Z} \sum_p \sum_q \frac{\partial Y^C}{\partial A_{pq}^k}, k = 1, \dots, K$
   // $K$ is the number of feature maps in the last layer of convolution layers
2: $L_X \leftarrow \text{ReLU}\left(\sum_k \lambda_k^C A^k\right)$
3: $mask_X = \begin{cases} 1 & [L_X] \geq t \\ 0 & \text{others} \end{cases}$   // Get FCRs
4: $\delta_1 = \epsilon_1 \times \text{sign}\left(\nabla_X J(\theta, X, y) \odot mask_X\right), \delta_2 = \epsilon_2 \times \text{sign}\left(\nabla_X J(\theta, X, y) \odot \overline{mask_X}\right)$
5: $X' = X + \delta_1 + \delta_2$
6: $X' = \text{clip}\left(X', \min, \max\right)$
7: **end for**

---

Table 3: The attack success rate of adding different perturbations to two different semantic parts.

| Model | $\epsilon_1$ (FCRs) | $\epsilon_1$ (Non-FCRs) | ASR |
|---|---|---|---|
| | 0.03 | 0.00 | 35.12% |
| | 0.03 | 0.01 | 36.43% |
| | 0.03 | 0.03 | 38.71% |
| VGG-11 | 0.03 | 0.05 | 40.34% |
| | 0.03 | 0.03 | 38.71% |
| | 0.05 | 0.03 | 48.65% |
| | 0.05 | 0.05 | 49.92% |
| | 0.08 | 0.05 | 59.62% |
| | 0.03 | 0.00 | 33.47% |
| | 0.03 | 0.01 | 34.17% |
| | 0.03 | 0.03 | 35.24% |
| ResNet-18 | 0.03 | 0.05 | 36.07% |
| | 0.03 | 0.00 | 33.47% |
| | 0.05 | 0.00 | 40.58% |
| | 0.05 | 0.03 | 42.23% |
| | 0.05 | 0.05 | 43.21% |

# B   ANALYSIS OF HYPER-PARAMETERS

**Iteration Times $N$ and Inverse Temperature $T$:** $N$ and $T$ are the dominant hyper-parameter in the proposed algorithm, and here we explore them effects on ASR. We can observe that both $N$ and $T$ have positive trends on the ASR (Figure 4(a) 4(b)). As $N$ and $T$ increase, the ASR also tends to increase. When $N = 30$, the ASR of the FCRs attack can reach 100% on both datasets. The ASR increases fastest when $N = 1$ to $N = 5$, and then it tends to grow slowly until 100%. With the increase of the iteration times, our objective function can better find the global optimal solution, thereby avoiding falling into the local optimal solution. The increase of $T$ also obviously leads to the high ASR because the increase of $T$ can make $J_{adv}^T$ and the regularization function $P$ become smaller, which makes our objective function $1/F$ continue to decrease and is better to find the optimal solution when performing stochastic gradient descent. It needs to be explained here that we find that the best situation can be achieved when $T = 0.05$, especially on the ILSVRC2012 dataset, but the attack effect may be reduced if $T$ continues to increase.

**Threshold $t$:** The threshold $t$ is also the dominant hyper-parameter which size directly determines the size of the $mask_X$, that is, the size of the range of adding perturbations. Specifically, we vary $t$ while keeping the other parameters fixing to observe the influence of $t$ changes on the ASR and

the $\ell_0$ norm of the perturbations. When $t = 0$, the ASR on both datasets reaches 100%. At the same time $\ell_0$ is 2903 and 198402, respectively. As the threshold $t$ continues to increase, the range of perturbations continue to decrease. The direct manifestation is that the norm decreases linearly. When reaching 0.5, the norms drop to 1529 and 24026 on the two datasets, respectively, which are 1/2 and 1/10 of $t = 0$. However, the ASR does not drop dramatically, which is reduced by 0.7% on the CIFAR-10 and 5.07% on the ILSVRC2012 (Figure 5(a) 5(b)). In the experiments of this paper, we set the threshold $t = 0.2$ on both datasets.

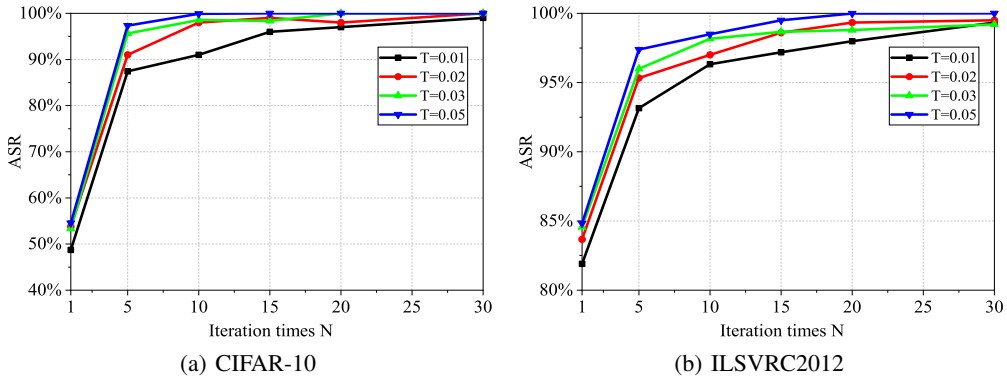

Figure 4: The effect of iteration times $N$ and hyper-parameter $T$ on the ASR. Using ResNet-18 network on the CIFAR-10 ($t = 0.2$, $LR = 10$, $\beta = 1$); using VGG-16 network on the ILSVRC2012 ($t = 0.2$, $LR = 20$, $\beta = 1$).

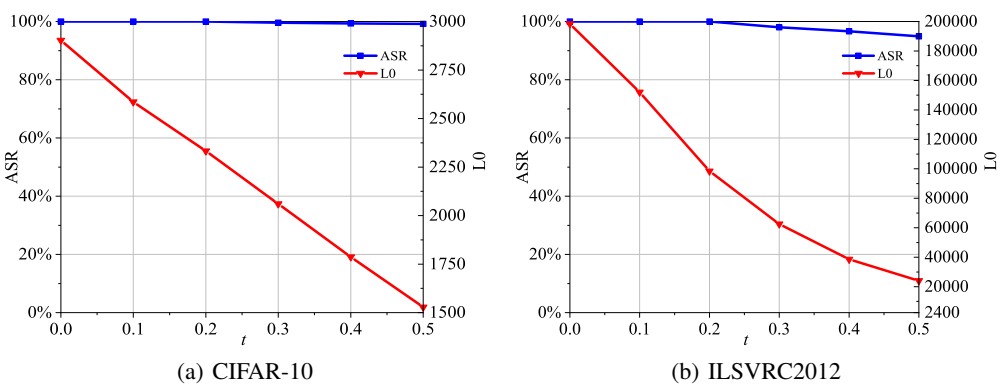

Figure 5: The effect of iteration times $N$ and hyper-parameter $T$ on the ASR. Using ResNet-18 network on the CIFAR-10 dataset ($t = 0.2$, $LR = 10$, $\beta = 1$); using VGG-16 network on the ILSVRC2012 dataset ($t = 0.2$, $LR = 20$, $\beta = 1$).

## C    THE RESULTS OF BLACK-BOX ATTACK

Table 4: ASR of alternative model and target model under various attack methods on the CIFAR-10. The diagonal blocks show the white-box attacks, while the off-diagonal blocks indicate black-box attacks which are much more challenging: PGD ( iteration number: $s = 20$, $\epsilon$-ball: $\epsilon = 16$, step size: $\alpha = 2$), M-DI$^2$-FGSM ($s = 20$, $\epsilon = 16$, $\alpha = 2$, the transformation probability: $p = 0.5$), C&W ($c = 1$, $lr = 0.01$, $iterations = 1000$), Ours ($t = 0.2$, $T = 0.03$, $LR = 10$, $N = 20$, $\beta = 1$).

| Model | Attack | VGG-16 | VGG-11 | VGG-13 | ResNet-18 | ResNet-34 |
|---|---|---|---|---|---|---|
| VGG-16 | PGD | 90.43% | 56.57% | 71.72% | 58.59% | 56.78% |
| | M-DI$^2$-FGSM | 90.79% | 66.78% | 83.94% | 70.84% | 69.38% |
| | C&W | 96.97% | 51.52% | 52.53% | 40.40% | 48.48% |
| | Ours | 96.00% | **81.88%** | **88.29%** | **75.00%** | **77.08%** |
| VGG-11 | PGD | 63.46% | 93.51% | 65.17% | 55.18% | 58.15% |
| | M-DI$^2$-FGSM | 90.11% | 99.10% | 90.69% | 87.79% | 85.09% |
| | C&W | 46.46% | 93.94% | 45.23% | 39.70% | 49.25% |
| | Ours | 86.97% | **99.90%** | **91.00%** | 86.12% | **85.69%** |
| VGG-13 | PGD | 75.08% | 61.33% | 95.81% | 61.36% | 58.91% |
| | M-DI$^2$-FGSM | 80.11% | 74.47% | 97.70% | 79.78% | 75.38% |
| | C&W | 64.11% | 62.13% | 95.72% | 61.95% | 61.98% |
| | Ours | **90.39%** | **79.78%** | **100.00%** | **80.38%** | **80.68%** |
| ResNet-18 | PGD | 60.92% | 60.70% | 62.52% | 93.18% | 64.22% |
| | M-DI$^2$-FGSM | 83.42% | 72.83% | 82.12% | 94.60% | 84.35% |
| | C&W | 54.88% | 61.28% | 55.97% | 97.44% | 56.24% |
| | Ours | **85.03%** | **85.09%** | **84.18%** | **100%** | **89.59%** |
| ResNet-34 | PGD | 60.92% | 60.28% | 64.56% | 68.47% | 92.72% |
| | M-DI$^2$-FGSM | 88.89% | 78.08% | 87.78% | 91.39% | 98.60% |
| | C&W | 46.00% | 57.75% | 47.40% | 48.59% | 90.34% |
| | Ours | **89.00%** | **84.95%** | 86.29% | 90.95% | **100%** |

Table 5: ASR of alternative model and target model under various attack methods on the ILSVRC2012. Parameter settings are as follows: PGD ( iteration number: $s = 20$, $\epsilon$-ball: $\epsilon = 16$, step size: $\alpha = 2$), M-DI$^2$-FGSM ($s = 20$, $\epsilon = 16$, $\alpha = 2$, the transformation probability: $p = 0.5$), C&W ($c = 1$, $lr = 0.01$, $iterations = 1000$), Ours ($t = 0.2$, $T = 0.05$, $LR = 20$, $N = 20$, $\beta = 1$).

| Model | Attack | VGG-16 | VGG-19 | ResNet-34 | ResNet-50 | ResNet-101 |
|---|---|---|---|---|---|---|
| VGG-16 | PGD | 99.40% | 89.00% | 71.70% | 70.30% | 62.70% |
| | M-DI$^2$-FGSM | 99.80% | 92.00% | 70.90% | 71.20% | 63.30% |
| | C&W | 97.00% | 85.90% | 71.70% | 70.70% | 68.70% |
| | Ours | **100.00%** | **92.90%** | **75.80%** | **76.80%** | **72.20%** |
| VGG-19 | PGD | 90.50% | 99.00% | 71.40% | 67.80% | 64.80% |
| | M-DI$^2$-FGSM | 96.00% | 99.00% | 67.70% | 72.60% | 64.50% |
| | C&W | 83.80% | 95.50% | 70.90% | 69.80% | 67.80% |
| | Ours | 94.20% | **99.20%** | **80.50%** | **75.60%** | **74.20%** |
| ResNet-34 | PGD | 82.60% | 82.60% | 99.40% | 76.20% | 67.70% |
| | M-DI$^2$-FGSM | 78.80% | 81.40% | **100.00%** | 86.60% | 81.30% |
| | C&W | 81.50% | 81.90% | 94.50% | 70.90% | 67.80% |
| | Ours | **89.20%** | **86.20%** | **100.00%** | **90.00%** | **87.10%** |
| ResNet-50 | PGD | 82.60% | 78.80% | 72.30% | 99.60% | 69.60% |
| | M-DI$^2$-FGSM | 77.40% | 77.90% | 79.80% | 99.50% | 87.90% |
| | C&W | 81.80% | 82.40% | 71.90% | 95.50% | 67.30% |
| | Ours | **87.60%** | **84.60%** | **91.00%** | **99.80%** | **89.60%** |
| ResNet-101 | PGD | 81.60% | 77.40% | 70.70% | 75.70% | 99.60% |
| | M-DI$^2$-FGSM | 72.70% | 69.30% | 77.80% | 88.90% | 99.70% |
| | C&W | 83.90% | 81.90% | 72.70% | 70.40% | 95.50% |
| | Ours | **86.30%** | **84.90%** | **90.00%** | **92.50%** | **100.00%** |

