# OpenReview forum: "An Adversarial Attack via Feature Contributive Regions"
_ICLR.cc/2021/Conference — Reject_

### Official Review · AnonReviewer1 · 2020-10-27
**Well written but incremental work**

**Rating:** 3
**Confidence:** 4

**Review:**

This paper focuses on the problem of generating sparse l2-adversarial examples in a white-box and surrogate/transfer setting. The authors consider “local attacks” – perturbing on a limited number of pixels while achieving high attack success rate. The main contribution of this work is to define the region to perturb using grad-cam based saliency maps to identify regions that have a greater impact on the classification decision. Having identified this region, the author use SGD to find the adversarial perturbations. The experimental results show that a high attack success rate can be achieved with this method.

Pros:
1)	The idea is simple yet affective. Attack success rate seems to be high.
2)	Paper is well written and easy to read.
3)	The experimental setup is reasonable.

Cons:
1) My main concern with this paper is that it appears incremental to me. The idea builds upon Grad-cam and existing adversarial example generation approaches.
2)	Also, the idea is not new implying that there exist several related approaches achieving sparse adversarial examples (see [1,2]).
3)	I would encourage authors to perform a more detailed literature survey and compare their results with them in a fair manner. The metric should not just be the attack success rate but also the number of perturbed pixels.
4)	The robustness analysis of adversarial examples to image transformations or adversarial example detection methods can be interesting.


[1] Kaidi Xu et al., STRUCTURED ADVERSARIAL ATTACK: TOWARDS GENERAL IMPLEMENTATION AND BETTER INTERPRETABILITY, ICLR 2019
[2] Hai Shu et. al., Adversarial Image Generation and Training for Deep Neural Networks

---

### Official Review · AnonReviewer2 · 2020-10-28
**The novelty of the paper is marginal and more experimental analysis is needed.**

**Rating:** 5
**Confidence:** 4

**Review:**

The paper proposes a method for adversarial attacking, which focuses on the contributive region of the input image.
However, there are several weaknesses:
1.	The interested-region-based adversarial attacking is not a novel idea, it has been proposed earlier:
Yao et al. Trust Region Based Adversarial Attack on Neural Networks, CVPR 2019.
2.	The proposed method in the paper is spliced by existing methods. The feature contributive regions are generated by Grad-CAM, as the authors mentioned in Section 3.2. The perturbation generation is also similar to general gradient-based methods except for the mask of the feature contributive regions.
3.	The way that the information of the contributive region is leveraged is somewhat crudity. A naïve binary mask is injected into the iteration of perturbation generation. More analysis is desired. For example, what if we use a soft mask of the contributive region, rather than a hard mask, for adversarial attacking?

---

### Official Review · AnonReviewer4 · 2020-11-04
**Comments on "An Adversarial Attack via Feature Contributive Regions"**

**Rating:** 3
**Confidence:** 2

**Review:**

In this paper, the authors proposed a new adversarial attack method based on feature contributive regions. The authors firstly utilize an off-the-shelf method to extract FCR of the image and then utilize the FCR as constraint for the perturbation. I tend to reject this paper due to following reasons:
1) The presentation of this paper needs significant improvement, the current manuscript is hard to read. Furthermore, the equations and many mathimatical presentations are unprofessional.
2) Although I am not familiar with the adversarial attack literature,  I think the method proposed in this paper is quite straightforward and the contribution of this paper is not significant enough for ICLR.
3) The FCR is estimated for the original input, while, introducing perturbations might change the FCR (but I guess the FCR won't change significantly). Instead of adopting a two-stage approach which utilize fixed FCR, have you thought about generating perturbation and update FCR jointly or alternatively?
4) I think the authors should analysis the effect of FCR threshold.

---

### Decision · Program_Chairs · 2021-01-07
**Final Decision**

**Decision:**

Reject

**Comment:**

This paper proposes an adversarial attack method based on feature contributive regions. In generating adversarial perturbations, Grad-CAM heatmaps are used as constraints for the perturbation. The overall idea is interesting and straightforward. However, as several reviewers raised, similar methods have been proposed in prior literature (Yao et al. CVPR 2019 and Xu et al. ICLR 2019), therefore making the novelty questionable.

All three reviewers have indicated rejection, and there is no author response. The paper is therefore rejected.